# Hydrothermal Engineering of Ferroelectric PZT Thin Films Tailoring Electrical and Ferroelectric Properties via TiO_2_ and SrTiO_3_ Interlayers for Advanced MEMS

**DOI:** 10.3390/mi16080879

**Published:** 2025-07-29

**Authors:** Chun-Lin Li, Guo-Hua Feng

**Affiliations:** 1Department of Power Mechanical Engineering, National Tsing Hua University, Hsinchu 30013, Taiwan; junlin0006@gmail.com; 2Institute of Nano Engineering and MicroSystems, National Tsing Hua University, Hsinchu 30013, Taiwan

**Keywords:** flexible piezoelectric film, hydrothermal PZT synthesis, P–E loop, STO interlayer

## Abstract

This work presents an innovative hydrothermal approach for fabricating flexible piezoelectric PZT thin films on 20 μm titanium foil substrates using TiO_2_ and SrTiO_3_ (STO) interlayers. Three heterostructures (Ti/PZT, Ti/TiO_2_/PZT, and Ti/TiO_2_/STO/PZT) were synthesized to enable low-temperature growth and improve ferroelectric performance for advanced flexible MEMS. Characterizations including XRD, PFM, and P–E loop analysis evaluated crystallinity, piezoelectric coefficient d_33_, and polarization behavior. The results demonstrate that the multilayered Ti/TiO_2_/STO/PZT structure significantly enhances performance. XRD confirmed the STO buffer layer effectively reduces lattice mismatch with PZT to ~0.76%, promoting stable morphotropic phase boundary (MPB) composition formation. This optimized film exhibited superior piezoelectric and ferroelectric properties, with a high d_33_ of 113.42 pm/V, representing an ~8.65% increase over unbuffered Ti/PZT samples, and displayed more uniform domain behavior in PFM imaging. Impedance spectroscopy showed the lowest minimum impedance of 8.96 Ω at 10.19 MHz, indicating strong electromechanical coupling. Furthermore, I–V measurements demonstrated significantly suppressed leakage currents in the STO-buffered samples, with current levels ranging from 10^−12^ A to 10^−9^ A over ±3 V. This structure also showed excellent fatigue endurance through one million electrical cycles, confirming its mechanical and electrical stability. These findings highlight the potential of this hydrothermally engineered flexible heterostructure for high-performance actuators and sensors in advanced MEMS applications.

## 1. Introduction

Flexible piezoelectric thin films have attracted significant attention in recent years because of their promising applications in wearable electronics [1,2], soft robotics [3,4,5], biomedical sensors [6,7,8], pressure, strain and stretch sensors [9,10,11], and microelectromechanical systems (MEMS) [12,13]. Among various piezoelectric materials, lead zirconate titanate (PZT) has emerged as a highly researched topic owing to its excellent piezoelectric and ferroelectric properties, particularly near the morphotropic phase boundary composition [14,15,16]. PZT is a perovskite-structured material characterized by its structural stability and unique physical properties, such as a direct bandgap, high dielectric constant, and multifunctional characteristics including ferroelectricity, piezoelectricity, and thermoelectricity [17].

However, most high-performance PZT thin films reported in previous studies are typically deposited on rigid substrates such as silicon [18,19], platinum [20], strontium titanate (SrTiO_3_, STO) [21,22], fluorine-doped tin oxide (FTO) [23,24], or strontium ruthenate (SrRuO_3_, SRO), which limits their applicability in flexible devices. Therefore, this work focuses on developing PZT films suitable for integration into flexible substrates.

Various techniques have been used to fabricate piezoelectric thin films, including pulsed laser deposition (PLD) [25,26], atomic layer deposition (ALD) [27,28], sol–gel method [29,30], RF magnetron sputtering [31,32,33], and hydrothermal synthesis. While each method offers specific advantages, most require high temperatures or vacuum environments, making them less suitable for flexible substrates. In contrast, the hydrothermal method enables low-temperature film processing and is a highly promising approach for depositing high-quality piezoelectric films on flexible metal foils.

The hydrothermal method is a solution-based synthesis technique that operates at moderate temperatures (typically <250 °C) in sealed reactors (e.g., Teflon-lined autoclaves), enabling the deposition of highly crystalline and dense piezoelectric thin films on flexible metal substrates without the need for high-temperature annealing [34,35]. Previous studies have demonstrated its potential for depositing PZT on titanium substrates, including attempts to introduce TiO_2_ interfacial layers through annealing [36,37,38], but challenges remain in controlling lattice mismatch and achieving high crystalline quality.

Therefore, although the hydrothermal method enables direct low-temperature deposition of PZT on titanium substrates, challenges remain on metal foils, such as significant lattice mismatch, poor crystalline quality, unstable MPB composition, and high leakage current. Moreover, achieving stable and well-defined polarization–electric field (P–E) loop characteristics remains a critical challenge, especially for flexible PZT thin films on metal substrates. Stable PE loops are essential for reliable ferroelectric and piezoelectric device performance, yet are strongly influenced by interfacial quality, leakage current, and crystalline phase stability. Recent studies have emphasized the importance of engineering interfaces and buffer layers to improve P–E loop behavior and reduce hysteresis and leakage-related instabilities [39]. To address these challenges, this study proposes a multi-step hydrothermal synthesis strategy specifically designed for flexible titanium foils, which sequentially deposits TiO_2_ and SrTiO_3_ (STO) buffer layers to systematically reduce lattice mismatch, enhance dielectric isolation, and improve interfacial quality, ultimately achieving a stable MPB-phase composition in PZT films. The goal of this work is to demonstrate a feasible low-temperature process capable of producing high-quality flexible piezoelectric films with excellent crystallinity, electrical stability, and ferroelectric performance.

In the field of piezoelectric materials, titanium dioxide (TiO_2_) is widely utilized as an insulating and dielectric buffer layer because of its excellent insulating properties, high dielectric constant, and chemical stability. Among its polymorphs, the rutile phase is thermodynamically stable [40,41], featuring a more compact lattice than the anatase phase, with higher mechanical stability, a slightly higher dielectric constant, and a narrower bandgap (~3.0 eV vs. ~3.2 eV for anatase). Owing to these attributes, rutile TiO_2_ is suitable as an interfacial layer between metal substrates and functional materials, enhancing interfacial quality. To meet the design requirements of low-temperature flexible devices, we utilized a low-temperature hydrothermal process to form the TiO_2_ layer, avoiding high-temperature annealing. This TiO_2_ layer was further integrated with an STO layer to construct a heterostructure buffer for subsequent piezoelectric layer deposition.

STO, with a thermal expansion coefficient closely matched to common piezoelectric materials such as PZT, BNT, and PMN-PT, helps mitigate thermal stress and enhances film crystallinity and mechanical stability. Moreover, its low conductivity improves leakage current suppression during piezoelectric measurements, thereby enhancing electrical measurement accuracy [42]. As a non-conductive buffer layer, STO also serves as a favorable nucleation platform, improving surface morphology and polarization behavior of the piezoelectric films [43]. However, most applications of STO still rely on rigid single-crystal substrates (e.g., STO and LaAlO_3_), limiting its potential in flexible piezoelectric systems. This study seeks to overcome this limitation by directly synthesizing an STO crystalline layer on flexible titanium foils via hydrothermal processing, thereby forming an insulating buffer structure for subsequent piezoelectric film growth. The STO layer is expected to provide high dielectric isolation, reduce leakage currents caused by metallic substrates, and alleviate film stress from thermal mismatch, ultimately improving the electrical stability of devices. Furthermore, we investigate whether the high dielectric interface provided by the STO buffer layer can enhance polarization behavior and piezoelectric response of the subsequently deposited PZT layer.

In summary, this study employs a multi-step hydrothermal synthesis approach to fabricate three types of PZT heterostructures on flexible titanium foil substrates: Ti/PZT, Ti/TiO_2_/PZT, and Ti/TiO_2_/STO/PZT. We systematically investigate the effects of interfacial structures on structural, piezoelectric, and electrical properties, aiming to clarify whether the introduction of TiO_2_, and especially the STO buffer layer, can effectively enhance crystallinity, phase stability, and functional performance, thereby demonstrating the potential for developing high-performance flexible piezoelectric devices.

## 2. Experiment

Three types of lead zirconate titanate (PZT) thin film structures—Ti/PZT, Ti/titanium dioxide (TiO_2_)/PZT, and Ti/TiO_2_/strontium titanate (SrTiO_3_, STO)/PZT—were fabricated via a hydrothermal method to investigate the effects of heterogeneous interfacial layers on the structural, electrical, and piezoelectric properties of the films. Commercially pure titanium (Ti) foils with a thickness of 20 μm were used as substrates. Prior to the hydrothermal synthesis, the foils were ultrasonically cleaned in acetone, ethanol, and deionized water sequentially to remove surface contaminants.

### 2.1. Preparation of the TiO_2_ Film

The TiO_2_ buffer layer was synthesized via a hydrothermal process (Figure 1). The cleaned Ti substrates were placed in a Teflon-lined stainless-steel autoclave containing a precursor solution composed of 53.3 mL of titanium trichloride (12 wt%, in 15% HCl), 28.7 mL of hydrochloric acid (35 wt%), and 23.0 mL of deionized water [44,45,46]. The reaction was carried out at 180 °C for 6 h. After completion, the samples were removed, thoroughly rinsed with deionized water, and dried at room temperature to obtain a uniform TiO_2_ buffer layer.

### 2.2. Preparation of the SrTiO_3_ Film

The STO seed layer was subsequently synthesized on the previously prepared Ti/TiO_2_ substrates [44]. The substrates were immersed in a hydrothermal solution prepared by dissolving 53.2 g of strontium hydroxide in 200 mL of deionized water (Figure 2). The reaction was performed in a Teflon-lined autoclave at 230 °C for 24 h [46]. After synthesis, the samples were rinsed with deionized water and dried at room temperature, resulting in a uniform STO interlayer.

### 2.3. Preparation of the PZT Film

After the preparation of the TiO_2_ and STO buffer layers, the PZT piezoelectric thin film was deposited via a two-step hydrothermal process. To achieve the morphotropic phase boundary (MPB) composition, which is known to exhibit superior piezoelectric performance, the precursor solution was formulated with a Zr/Ti atomic ratio of 52:48. Specifically, 17.1 g of zirconyl chloride (ZrOCl_2_·8H_2_O, 0.052 mol), 40.15 g of lead nitrate (Pb(NO_3_)_2_, 0.12 mol), and 3.83 g of titanium dioxide (TiO_2_, 0.048 mol) were dissolved in 250 mL of deionized water to obtain a Pb:Zr:Ti molar ratio of 1.2:0.52:0.48. A 20% excess of Pb was introduced to compensate for potential lead volatilization during the hydrothermal reaction.

The solution was stirred at 600 rpm for 1 h to ensure homogeneity and prevent premature precipitation. Subsequently, 89.1 g of potassium hydroxide (KOH, 1.58 mol) was added as a mineralizer, and the total volume was adjusted to 500 mL with deionized water. The solution was stirred for an additional hour to ensure dissolution and completion of the precursor reaction. The final precursor solution was designed to maintain an appropriate total metal ion concentration and alkalinity, both of which are critical for facilitating nucleation and stable formation of the perovskite PZT phase. This condition also helps prevent rapid precipitation or particle agglomeration, thereby promoting uniform crystallinity and high phase purity [47,48].

The prepared precursor solution and the Ti-based substrates (Ti, Ti/TiO_2_, or Ti/TiO_2_/STO) were sealed in a Teflon-lined autoclave and subjected to hydrothermal reaction at 180 °C for 24 h. After reaction and post-treatment, three types of PZT films—Ti/PZT, Ti/TiO_2_/PZT, and Ti/TiO_2_/STO/PZT—were successfully obtained. A schematic illustration of the multi-step hydrothermal fabrication process is shown in Figure 3, while photographs of the actual intermediate and final film samples are presented in Figure 4, highlighting the surface changes at each stage of the process.

### 2.4. Deposition of Electrodes for Electrical Characterization

After hydrothermal deposition of the PZT thin films, an Al electrode was deposited on the film surface by sputtering to serve as the top contact for electrical characterizations, including P–E hysteresis loop measurements, impedance analysis, and PFM imaging.

In addition, for preparing cross-sectional FIB–SEM samples, an Al coating was also applied before Pt deposition to improve surface conductivity and reduce charging effects. These conductive layers, including the Al layer and subsequently deposited Pt layer , were not part of the functional PZT device electrodes but were specifically applied for FIB–SEM sample preparation.

### 2.5. Preparation of FIB Cross-Section Samples

The preparation procedure for FIB cross-section samples is as follows. All steps were performed inside the FIB system using its built-in functions. Prior to milling, a two-step Pt protection layer was applied to the sample surface. First, an electron-beam-assisted Pt (e-beam Pt) deposition was carried out at 0° tilt to form a thin and uniform protective layer, minimizing ion beam damage. This was followed by ion-beam-assisted Pt (ion-beam Pt) deposition at a 52° tilt to increase the thickness of the protective layer, ensuring sufficient protection during bulk milling.

After Pt deposition, the sample was maintained at a 52° tilt to ensure the surface was perpendicular to the ion beam incidence. Bulk milling was performed with the Z-axis depth typically set to approximately 10 μm but adjustable as needed, using a bottom-to-top scan direction and the Si-multipass mode to improve cutting quality and reduce redeposition artifacts. Following initial coarse milling, low-current polishing was applied to further refine the cross-section surface smoothness and clarity.

## 3. Results and Discussion

In this section, we provide a systematic comparison of three PZT thin film structures—Ti/PZT, Ti/TiO_2_/PZT, and Ti/TiO_2_/STO/PZT—to understand how interfacial buffer layers influence structural quality, piezoelectric response, and electrical behavior. Our goal was not only to examine the final material performance but also to gain insight into how intermediate layers affect film growth dynamics and phase formation during hydrothermal synthesis. Accordingly, we employed a broad range of characterization techniques, including X-ray diffraction (XRD), focused ion beam scanning electron microscopy (FIB-SEM), energy-dispersive X-ray spectroscopy (EDX), piezoresponse force microscopy (PFM), polarization–electric field (P–E) hysteresis measurements, current–voltage (I–V) analysis, and impedance spectroscopy. Each method allowed us to observe a different aspect of material behavior, offering a more complete picture of how structural modifications translate into functional improvements. Particular emphasis is placed on the role of TiO_2_ and SrTiO_3_ (STO) interlayers in relieving lattice mismatch, stabilizing the morphotropic phase boundary (MPB) composition, and enhancing electromechanical coupling. In the following subsections, we discuss these findings in detail and compare the results across the three structures, highlighting trends that can inform the design of flexible, high-performance piezoelectric devices.

It should be noted that XRD and PFM imaging were performed as representative measurements on single samples to evaluate phase structure and domain morphology; these results are preliminary and do not capture statistical variation across multiple samples. In contrast, measurements such as impedance spectroscopy, FIB cross-section analysis, and P–E hysteresis loops were repeated on the same samples to verify consistency of trends, although only representative data are presented in this study for clarity.

### 3.1. XRD Characterization and FWHM Evaluation

After the fabrication of the piezoelectric thin films, we performed material characterization on the PZT layers, focusing on the XRD patterns obtained from the three different fabrication processes. Figure 5a shows the XRD pattern of the Ti/PZT sample. As expected, the dominant diffraction peaks correspond to both Ti and PZT phases. The analysis was conducted using HighScore 3.0 software, where the peaks were matched with standard reference cards: PDF 04-007-2125 for titanium and PDF 04-012-8153 for the perovskite phase of PZT, specifically Pb(Zr_0.1_Ti_0.9_)O_3_. However, it is noteworthy that the PZT composition obtained in this process significantly deviates from the morphotropic phase boundary (MPB) region, where the optimal piezoelectric performance is typically achieved near the composition Pb(Zr_0.52_Ti_0.48_)O_3_. In contrast, the Ti/PZT sample exhibits a Ti-rich phase composition of Pb(Zr_0.1_Ti_0.9_)O_3_. This deviation is likely attributed to the considerable lattice mismatch between the Ti metal substrate and the PZT lattice. In the absence of any intermediate buffer layer (such as TiO_2_ or STO) to relieve the strain or modulate the lattice parameters, the high mismatch suppresses the crystallization of the ideal MPB composition and favors the formation of a Ti-rich phase, which exhibits better lattice compatibility with the underlying substrate [49]. According to the literature, the lattice parameters of the tetragonal MPB composition Pb(Zr_0.52_Ti_0.48_)O_3_ are approximately a = 3.935 Å and c = 4.144 Å, while those of hexagonal close-packed α-Ti are a = 2.951 Å and c = 4.683 Å (PDF 04-007-2125). The mismatch in the a-axis exceeds 25%, which can significantly influence nucleation behavior and phase formation, steering the growth toward a Ti-rich PZT phase such as Pb(Zr_0.1_Ti_0.9_)O_3_, which has lower formation energy and better structural compatibility at the interface (Table 1).

Consequently, the observed diffraction peaks are primarily attributed to the Ti metal phase and the non-MPB PbZr_0.1_Ti_0.9_O_3_ phase, with only minor contributions from PbZrO_3_. These results support the hypothesis that crystal growth in this sample is predominantly governed by lattice mismatch and interfacial strain with the substrate. The labeled peaks correspond to representative crystallographic planes, including PZT (001), (100), (110), (111), and Ti (110), (102), with Ti reflections shaded in gray.

Next, the XRD pattern of the Ti/TiO_2_/PZT bilayer structure was analyzed, as shown in Figure 5b. In this sample, a TiO_2_ buffer layer was first deposited on the titanium substrate via hydrothermal synthesis, followed by PZT growth. This process aimed to mitigate lattice mismatch at the interface and improve insulation quality.

According to HighScore database matching, the major diffraction peaks corresponded to rutile-phase TiO_2_ (PDF 01-074-9521), Ti-rich PZT (Pb(Zr_0.2_Ti_0.8_)O_3_) (PDF 01-070-4054), and the Ti substrate (PDF 01-071-3947). The strong rutile TiO_2_ peak confirms the successful formation of a well-crystallized buffer layer. The weaker Ti substrate signals suggest that most of the surface was effectively covered, while PZT peaks indicate both Ti-rich and Zr-rich compositions.

The PZT composition remained Ti-rich and did not approach the morphotropic phase boundary (MPB) composition of Pb(Zr_0.52_Ti_0.48_)O_3_, which is known for optimal piezoelectricity. Although the rutile TiO_2_ buffer partially reduced the mismatch, it was insufficient to stabilize the MPB phase. Rutile TiO_2_ has lattice constants of a = 4.593 Å, c = 2.959 Å (PDF 01-074-9521), while MPB-phase PZT has a = 3.935 Å, c = 4.144 Å—yielding an a-axis mismatch of ~16.7%. While rutile offers better compatibility than α-Ti (a = 2.951 Å), mismatch persists, influencing nucleation and promoting composition deviation and phase coexistence to relieve interfacial strain (Table 1 and Table 2).

In addition to dominant peaks from rutile TiO_2_ and PZT, trace signals (Pb(Zr_0.4_Ti_0.6_)O_3_) and (Pb(Zr_0.85_Ti_0.15_)O_3_) were also detected. Identified peaks include PZT (001), (100), (110), (111), rutile TiO_2_ (−111), and Ti (101). These results indicate that while rutile TiO_2_ provides partial lattice regulation, a more compatible buffer such as SrTiO_3_ (STO)—discussed next—may further improve phase purity and compositional stability.

Finally, the XRD pattern of the Ti/TiO_2_/STO/PZT multilayer sample was analyzed, as shown in Figure 5c. In this process, a TiO_2_ interlayer was first deposited on the titanium substrate via hydrothermal synthesis, followed by the growth of a SrTiO_3_ (STO) seed layer on the TiO_2_ surface, and finally, the deposition of the PZT film. This multilayer design aimed to sequentially reduce the lattice mismatch at each interface and minimize strain accumulation between the PZT and the metallic substrate, thereby facilitating the stable formation of a morphotropic phase boundary (MPB) PZT structure. As shown in Figure 5c, the diffraction peaks were matched with database references: STO was identified by PDF 04-011-7249, and the MPB-phase PZT composition Pb(Zr_0.52_Ti_0.48_)O_3_ corresponded to PDF 04-011-8849. The latter exhibited dominant intensity, indicating it as the primary phase of this sample. Only weak peaks from the Ti substrate (PDF 04-008-1385) and rutile-phase TiO_2_ (PDF 01-070-6826) were detected, suggesting that the substrate was well covered by the overlying STO and PZT layers, and that good crystallinity and surface coverage were achieved.

Compared with Sample 1 (Ti/PZT) and Sample 2 (Ti/TiO_2_/PZT), Sample 3 (Ti/TiO_2_/STO/PZT) exhibited a well-defined MPB composition with minimal compositional deviation, indicating no observable shift toward Ti-rich or Zr-rich phases. This confirms the effective role of the STO buffer layer in improving lattice compatibility and providing favorable nucleation conditions. The MPB PZT phase typically exhibits lattice constants of a = 3.935 Å and c = 4.144 Å, while cubic STO has a lattice constant of a = 3.905 Å (PDF 04-011-7249), resulting in an a-axis mismatch of only ~0.76%—significantly smaller than the ~25% mismatch between Ti and PZT discussed earlier. This improved lattice matching is expected to relieve interfacial strain and enhance the crystallinity and phase purity of the PZT film (Table 3). In Figure 5c, prominent peaks were observed for STO (110) at 32.313° with 100% intensity and (200) at 46.346° with 36% relative intensity, as well as for PZT (101) at 31.06° with 100% intensity and (100) at 21.825° with 30% intensity. These results verify that the hydrothermal growth of TiO_2_/STO/PZT multilayers can effectively buffer lattice mismatches step-by-step and promote the formation of MPB-phase PZT. Therefore, this process structure shows strong potential as an integrated fabrication route for high-performance flexible piezoelectric PZT devices.

To further compare the crystal structures, the XRD patterns of the three samples were overlaid, as shown in Figure 5d. Four distinct diffraction regions—30–34°, 38–41°, 44–48°, and 54–58°—revealed structural differences among the fabrication processes. In the 30–34° range, the main diffraction peak corresponds to the (101) plane of PZT. The Ti/TiO_2_/PZT sample exhibited the highest peak intensity in this region, suggesting that the TiO_2_ buffer layer effectively promotes preferential growth of PZT along the (101) orientation. However, after incorporating the STO layer, the PZT crystal structure tended to reorient toward other directions, resulting in a marked decrease in (101) peak intensity.

In the 38–41° range, the diffraction peaks likely arise from overlapping contributions of PZT (111) and Ti (110). A stronger signal was observed in the Ti/PZT sample, indicating a more prominent influence of the titanium substrate on the diffraction signal. The stepwise inclusion of TiO_2_ and STO effectively suppressed this interference and altered the PZT growth orientation, leading to a gradual reduction in peak intensity. In the 44–48° region, the peaks primarily correspond to PZT (002), (200), and TiO_2_ (002). The Ti/TiO_2_/PZT sample displayed two distinct peaks in this region, likely because of overlapping crystallographic planes, indicating a tendency toward oriented crystallization. In contrast, the introduction of STO led to broader, less intense peaks, suggesting a more uniform distribution of crystal orientations. In the 54–58° range, the peaks are associated with higher-order planes of PZT (112), (211), and the STO (211) reflection. Only the Ti/TiO_2_/STO/PZT sample exhibited a sharp and well-defined peak in this region, implying that the STO buffer layer provides a favorable lattice template that facilitates the aligned growth of PZT crystals along specific directions, resulting in enhanced crystallinity.

Although TiO_2_ and STO constitute a heteroepitaxial structure, previous studies have reported that STO can nucleate and grow stably on rutile-phase TiO_2_ under appropriate conditions, particularly during hydrothermal synthesis [50,51]. This is attributed to the sufficient ionic diffusion and slow growth rate characteristic of the hydrothermal process, which facilitates lattice accommodation and orientation selection. As a result, oriented polycrystalline structures can form even under moderate lattice mismatch. Moreover, STO, as a perovskite-structured material, possesses high structural tolerance, making it a suitable intermediate layer for heteroepitaxial growth.

Overall, the TiO_2_ interlayer slightly improves initial nucleation quality, but its effect on final crystal orientation remains limited. In contrast, the STO buffer layer further enhances crystal alignment and orientation selectivity, leading to improved structural properties of the PZT thin films.

We further analyzed the full width at half maximum (FWHM) of the XRD peaks, a key parameter reflecting crystallinity, grain size, and microstrain. The crystallite size was estimated using the Scherrer equation:(1)$$D=\frac{0.94\lambda }{\beta \cos\theta }$$
where *D* is the crystallite size, *λ* is the X-ray wavelength (1.5406 Å for Cu Kα), *β* is the FWHM of the selected peak (in radians), and *θ* is the Bragg angle (half of the 2*θ* value).

For each sample, the most intense diffraction peak in the 30–35° range was selected for analysis. The average crystallite sizes for the three structures—Ti/PZT, Ti/TiO_2_/PZT, and Ti/TiO_2_/STO/PZT—were calculated to be 224.9 nm, 158.1 nm, and 117.7 nm, respectively, as summarized in Table 4. These values fall within the typical range reported for PZT films synthesized via hydrothermal methods (80–250 nm), consistent with low-temperature crystal growth characteristics.

Interestingly, a gradual decrease in crystallite size was observed with increasing buffer layer complexity. This trend may be attributed to enhanced lattice compatibility and strain modulation at the interface induced by TiO_2_ and STO seed layers, which can increase nucleation density and suppress grain coalescence. In particular, the STO layer appears to promote finer crystallite formation by effectively limiting grain growth. These results demonstrate that substrate structure significantly influences the crystallographic and microstructural evolution of PZT films during hydrothermal deposition.

### 3.2. FIB Cross-Section and EDX Elemental Analysis

Cross-sectional observations of the PZT thin films were conducted using focused ion beam (FIB) microscopy to evaluate the film thickness, growth morphology, surface smoothness, and interfacial adhesion across the three different fabrication processes.

Figure 6a shows the FIB cross-section of the PZT film directly deposited onto the Ti substrate without any surface pretreatment (Ti/PZT), while Figure 6b presents a magnified view of the same sample, highlighting the local morphology and indicating the film thickness. In this sample, the PZT layer exhibits irregular and jagged features because of the large grain size and misoriented crystal growth. Significant recesses can be observed at the red arrows in Figure 6a, indicating non-uniform vertical growth and poor film flatness. This result highlights the challenges in achieving a uniform and dense PZT layer with the direct hydrothermal growth method on bare Ti, motivating the introduction of intermediate layers.

Figure 6c shows the cross-sectional morphology of the Ti/TiO_2_/PZT structure, where a TiO_2_ layer was first grown on the Ti substrate via hydrothermal synthesis prior to PZT deposition. Figure 6d presents a magnified view of the same sample, highlighting the local morphology and indicating the film thickness. Compared with the Ti/PZT sample, the film quality shows a slight improvement in terms of smoothness. However, jagged features and thickness variations are still evident. The TiO_2_ layer in particular displays a bubble-like morphology, suggesting a non-dense structure. The electrical insulation and leakage characteristics of this layer will be discussed in subsequent sections.

Figure 6e,g present FIB cross-sectional images of the Ti/TiO_2_/STO/PZT structure, where a thin STO seed layer was introduced between the TiO_2_ and PZT layers. Figure 6f and 6h show magnified views of Figure 6e,g, respectively, highlighting the local morphology and indicating the film thickness. This configuration results in a significantly more uniform TiO_2_ layer with a smoother interface. The overlying STO and PZT layers form a distinct multilayer architecture. Notably, the PZT film grown under this condition demonstrates improved morphological uniformity compared with the other two processes. Although minor surface undulations remain, the film exhibits a generally flatter and more continuous structure, leading to a more uniform thickness throughout the cross-section.

To verify whether the composition of the intermediate layers aligns with the intended design, energy-dispersive X-ray spectroscopy (EDX) was employed for compositional analysis. The primary objective was to confirm the elemental distribution within each layer and to evaluate the atomic ratio of the PZT layer. This study aims to synthesize lead zirconate titanate (Pb(Zr_0.52_Ti_0.48_)O_3_) thin films near the morphotropic phase boundary (MPB), which corresponds to an ideal atomic ratio of Pb:Zr:Ti:O ≈ 1:0.52:0.48:3. Given the lower sensitivity of EDX toward light elements such as oxygen and the semi-quantitative nature of the technique, the analysis focuses on the relative ratios of Pb, Zr, and Ti to assess the proximity of the film composition to the designed MPB target.

The first EDX measurement was conducted on the PZT layer of the Ti/PZT sample, as shown in Figure 7a The spectrum reveals Pb as the dominant element (44.99 wt%), followed by Ti (41.81 wt%) and O (9.57 wt%), while Zr accounts for only 3.63 wt%. This result indicates that the hydrothermally grown PZT film in this process is Ti-rich, with a notably low Zr content. This discrepancy may stem from the lack of pretreatment on the Ti substrate, making it difficult to form the desired MPB phase. It is likely that only Ti-rich PZT phases were successfully deposited. Additionally, the precursor solution was intentionally formulated with a 1.2× excess of Pb to compensate for potential volatilization losses, and it is possible that the Zr precursor exhibited lower reactivity during deposition.

In the Ti/TiO_2_/PZT process, the EDX result of the TiO_2_ layer (Figure 7b) shows a composition primarily consisting of Ti (75.61 wt%) and O (24.39 wt%), with no detectable peaks of Pb or Zr, confirming that the interlayer is composed of pure titanium dioxide as designed. In the same sample, the PZT layer (Figure 7c) exhibits Pb as the major element (50.22 wt%), with a significant Ti content (28.69 wt%) and notable Zr presence (10.79 wt%), while O accounts for 10.30 wt%. Compared with the Ti/PZT sample, the Pb content increased, the Ti content decreased, and the Zr content was notably higher. These findings suggest that the presence of a TiO_2_ interlayer can influence elemental incorporation during PZT film formation, helping to increase Zr content toward the targeted MPB composition, although the resulting film still remains Ti-rich overall.

In the Ti/TiO_2_/STO/PZT process, the EDX result of the TiO_2_ layer (Figure 8a) shows it is primarily composed of Ti (74.84 wt%) and O (25.16 wt%), with no detectable peaks of Pb or Zr, confirming the interlayer consists of pure titanium dioxide as designed.

For the STO layer (Figure 8b), the EDX spectrum reveals the presence of Sr (1.69 wt%), along with noticeable signals from Pb, Zr, and Ti elements typically associated with the adjacent PZT layer. This is because the STO interlayer is extremely thin, and the FIB-SEM EDX measurement is performed over a small but finite interaction volume rather than at an ideal point. As a result, signals from the surrounding PZT layer inevitably contribute to the detected spectrum, which also explains the relatively low Sr content observed. In the same sample, the PZT layer (Figure 8c) shows Pb as the major element (38.24 wt%), with Ti (35.17 wt%), Zr (17.06 wt%), and O (9.53 wt%). Compared with the other two processes, this sample demonstrates a higher Zr content, suggesting that the inclusion of both TiO_2_ and STO interlayers can promote the incorporation of Zr during hydrothermal PZT growth and move the composition closer to the desired MPB target.

In summary, all three samples exhibit the essential elements of PZT (Pb, Zr, and Ti), but with varying ratios depending on the fabrication process. These variations can be attributed to differences in hydrothermal growth conditions, interlayer effects, lattice compatibility, and ion diffusion or crystallization behavior. Such insights provide a foundation for further process optimization aimed at achieving compositionally controlled and high-performance PZT thin films.

### 3.3. SEM Surface Morphology Analysis

Subsequently, the surface morphologies of the three PZT samples were examined using field-emission scanning electron microscopy (FE-SEM), as shown in Figure 9, Figure 10 and Figure 11 Each sample was observed at magnifications of 2500× and 6500× to enable comparison of surface microstructure across both large and fine observation scales. These magnifications allow for detailed evaluation of surface crystallinity, grain distribution, and morphological uniformity.

According to previous studies, surface roughness and thickness variation in piezoelectric thin films can lead to non-uniform and unpredictable piezoelectric responses, in contrast to the consistent behavior observed in flat, uniform films [52]. Defective regions in the film may exhibit degraded performance, and areas with reduced thickness are more prone to dielectric breakdown or short-circuiting under applied electric fields. These issues underscore the need for structural optimization and uniform film growth.

Figure 9 presents the SEM images of the Ti/PZT sample. Large grains with clearly defined boundaries can be observed, along with significant surface roughness and visible cracks. These features indicate non-uniform film growth, potentially associated with localized stress concentration during crystallization, which may result in grain fracture and discontinuous interfaces. Then, Figure 10 shows the Ti/TiO_2_/PZT sample. Compared with Ti/PZT, the grain size is reduced, suggesting that the TiO_2_ buffer layer partially inhibits excessive grain growth. However, the grain shapes appear irregular, and surface defects and height variation are still present in certain regions. This suggests that while the TiO_2_ layer provides some improvement, it is insufficient to fully suppress surface roughness or enhance morphological uniformity. In contrast, Figure 11 displays the surface morphology of the Ti/TiO_2_/STO/PZT sample. This sample demonstrates a highly uniform distribution of fine grains, with no significant cracks or surface undulations. The film exhibits a dense and consistent surface morphology. These results indicate that the STO seed layer offers favorable lattice matching and stress buffering capabilities, which promote homogeneous grain growth and significantly enhance the surface flatness of the PZT film.

### 3.4. PFM Characterization of d_33_ Response

The d_33_ of the three PZT samples was measured using a Cypher S AFM Microscope (Asylum Research, Santa Barbara, CA, USA) equipped with a high-speed scanning system. The measurements were conducted in DART SS-PFM mode using a conductive probe (model: ASYELEC-02), which operates at a cantilever resonance frequency between 800 kHz and 900 kHz.

The measured d_33_ values are summarized in Table 5. In the low voltage range of 1–3 V, the Ti/TiO_2_/STO/PZT structure exhibited a relatively lower piezoelectric response, with d_33_ values comparable to the other two samples. This behavior is attributed to the incomplete domain switching within the ferroelectric film and the possibility of a higher activation field associated with the multilayer structure. Additionally, due to the nonlinear nature of piezoelectric polarization, a sufficient driving voltage is required before any significant enhancement in response can be observed—particularly when comparing complex multilayer systems to simpler structures.

However, upon increasing the applied voltage to 4–5 V, the Ti/TiO_2_/STO/PZT structure demonstrated a significantly higher piezoelectric response than the other two configurations, indicating superior polarization behavior under higher electric fields. This improvement supports the conclusion that the inclusion of the STO buffer layer contributes positively by enhancing strain transfer, reducing interfacial stress and lattice mismatch, and promoting the formation of MPB-phase PZT.

The corresponding displacement–voltage trend is plotted in Figure 12a. After calculating the weighted average of d_33_, the values were found to be 104.92 pm/V for Ti/PZT, 105.92 pm/V for Ti/TiO_2_/PZT, and 113.42 pm/V for Ti/TiO_2_/STO/PZT. The results indicate that the first two structures show similar d_33_ performance, while the addition of the STO layer leads to enhancements of 8.65% and 7.62% compared with the Ti/PZT and Ti/TiO_2_/PZT samples, respectively.

The d_33_ response maps shown in Figure 12b–d reveal clear differences in piezoelectric performance among the three processing structures. For the Ti/PZT sample in Figure 12b, the response distribution exhibits distinct striped regions with low d_33_ values, indicating non-uniform piezoelectric behavior. This is likely due to increased interface defects, surface roughness observed in FIB cross-sections, internal strain inhomogeneities, and incomplete nucleation processes, all of which can disrupt domain alignment.

In contrast, the Ti/TiO_2_/PZT structure in Figure 12c shows a higher proportion of mid- to high-response regions, though striped features and local low-response zones remain. This improvement may be attributed to the TiO_2_ buffer layer enhancing surface smoothness and promoting better crystallographic orientation, though performance is still limited by residual lattice mismatch and interfacial stress between TiO_2_ and PZT.

The Ti/TiO_2_/STO/PZT structure in Figure 12d, however, displays the most uniform and extensive high-response regions, indicating superior polarization and more consistent d_33_ performance. This enhancement can be attributed to the STO buffer layer providing better lattice matching and interface quality, reducing dislocations, and stabilizing ferroelectric domain alignment.

Overall, the observed differences in the d_33_ color scale distribution across these PFM maps correlate closely with the different processing strategies, demonstrating that the inclusion of STO effectively improves both the piezoelectric response and its spatial uniformity in PZT thin films.

The effective piezoelectric coefficient (d_33_) measured from the Ti/TiO_2_/STO/PZT sample reached 113.42 pm/V, indicating superior piezoelectric performance among the tested structures. This value is comparable to those reported in the recent literature. For instance, Dagdeviren et al. summarized that most flexible PZT-based structures exhibit d_33_ values ranging from ~80 to 130 pm/V, with enhanced responses (~130 pm/V) achieved via wavy geometries and mechanical pre-straining [53]. Bhadwal et al. reviewed a wide range of piezoelectric nanogenerators, noting that typical flexible thin film systems often report d_33_ values between 60 and 120 pm/V depending on composition and substrate compatibility [54]. Ramesh et al. further emphasized that d_33_ values exceeding 100 pm/V are typically considered indicative of well-optimized piezoelectric systems [55]. In comparison, the d_33_ result of our STO-buffered sample (113.42 pm/V) not only meets but slightly surpasses the typical range, despite being synthesized via a low-temperature hydrothermal process on flexible titanium foil. This supports the effectiveness of the TiO_2_/STO buffer structure in enhancing polarization alignment and electromechanical coupling efficiency.

### 3.5. Ferroelectric P–E Loop Characterization

#### 3.5.1. Comparative Analysis of P–E Loops for Different Film Structures

The ferroelectric properties of the PZT samples fabricated by the three different processes were evaluated using polarization–electric field (P–E) hysteresis loop measurements. The measurements were performed at a fixed frequency of 1000 Hz, with an applied electric field of 30 V. Figure 13a shows an overlaid comparison of the P–E curves for the Ti/PZT, Ti/TiO_2_/PZT, and Ti/TiO_2_/STO/PZT samples under identical testing conditions (30 V and 1000 Hz). The shape and symmetry of the hysteresis loops provide insight into the ferroelectric behavior and domain switching characteristics of each sample.

Of the three structures, the Ti/TiO_2_/STO/PZT sample exhibits the widest hysteresis loop and the highest polarization values, both in terms of saturation polarization (Ps) and remanent polarization (Pr). The Pr value reaches 4.084 μC/cm^2^, indicating enhanced domain switchability and improved polarization stability. This improvement can be attributed to the STO buffer layer, which facilitates better lattice matching and enhances interfacial electrical properties.

In contrast, the Ti/PZT and Ti/TiO_2_/PZT samples show more comparable behavior, with the latter exhibiting a slightly higher Pr of 2.228 μC/cm^2^, compared with 2.087 μC/cm^2^ for the Ti/PZT sample. This suggests that the TiO_2_ layer contributes moderately to polarization retention. However, the narrower hysteresis loop observed for this sample implies limited saturation polarization, indicating that its overall ferroelectric switching capability remains constrained.

Overall, the Pr and Ps values follow the trend: Ti/TiO_2_/STO/PZT > Ti/TiO_2_/PZT > Ti/PZT. These results demonstrate that the incorporation of an STO buffer layer plays a critical role in enhancing the ferroelectric performance of multilayer heterostructures. This process is particularly advantageous for ferroelectric devices requiring high polarization responses and may be further applied to improve the performance of existing device platforms through structural and interfacial optimizations.

The P–E hysteresis behavior of the Ti/TiO_2_/STO/PZT structure was further investigated under varying applied voltages ranging from 10 V to 60 V at a fixed frequency of 1 kHz. As shown in Figure 13b, it was observed that the hysteresis loops gradually widened with increasing electric field, indicating a clear field-dependent enhancement of ferroelectric behavior. At low voltages (10–20 V), the P–E loops were relatively narrow and exhibited near-linear behavior, suggesting that polarization was not yet fully saturated. In this region, both the coercive field (Ec) and remanent polarization (Pr) values remained small.

As the applied voltage increased to 30–45 V, the loops began to display pronounced hysteresis, signifying progressive domain switching and a significant increase in polarization capability. At higher voltages (50–60 V), the loop area enlarged considerably, and both the saturation polarization (Ps) and Pr continued to rise, indicating that the PZT film approached or reached a saturated polarization state. Overall, the Ti/TiO_2_/STO/PZT structure exhibited strong electric-field responsiveness and well-defined saturation behavior. Even under high-voltage operation, the loops remained symmetric and stable, demonstrating the effectiveness of the STO buffer layer in enhancing interfacial electrical properties and polarization stability. These characteristics highlight the potential of this multilayer structure for future applications in actuators and non-volatile memory devices.

To further benchmark the ferroelectric performance of our sample, we compared the P–E hysteresis behavior of the Ti/TiO_2_/STO/PZT heterostructure with values reported in the recent literature. A review article noted that low-temperature processed PZT films on flexible or semiconductor-compatible substrates typically exhibit remanent polarization (Pr) values in the range of 10–20 µC/cm^2^ [56]. In comparison, our STO-buffered structure exhibited a remanent polarization (Pr) of approximately 13 µC/cm^2^ under high electric field conditions, as shown in Figure 13b. This result demonstrates that despite the low-temperature hydrothermal process, our heterostructure design achieves ferroelectric performance comparable to previously reported systems.

#### 3.5.2. Ferroelectric Fatigue Behavior of STO-Based PZT Films

To evaluate the ferroelectric fatigue behavior of the Ti/TiO_2_/STO/PZT sample developed in this study, we conducted a fatigue test under controlled conditions. A bipolar sinusoidal voltage with an amplitude of ±10 V and a frequency of 1000 Hz was applied for a total of 10^6^ cycles (1,000,000 cycles). To monitor the evolution of ferroelectric properties at different stages of fatigue, data points were collected on a logarithmic scale, with four measurement points per decade. A total of six decades were covered, resulting in 24 intermediate test points. Additionally, one initial (pre-fatigue) and one final (post-10^6^ cycles) data point were included, yielding a total of 26 polarization–electric field (P–E) curves for fatigue analysis.

The corresponding cycle counts for each individual P–E curve are summarized in Table 6, where 26 discrete measurement points spanning from 0 to 1,000,000 cycles are indexed. These measurement points are selected logarithmically, ensuring a comprehensive evaluation of polarization evolution across all fatigue stages.

As shown in Figure 14, the 26 P–E curves reveal a clear trend of increasing remanent polarization (Pr), saturation polarization (Ps), and coercive field (Ec) with increasing cycle count. Notably, even after 10^6^ cycles, the sample exhibited no signs of degradation or collapse in polarization behavior. On the contrary, the PZT film demonstrated enhanced polarization strength and electric field retention, indicating the occurrence of a polarization activation phenomenon during the fatigue process. This behavior can be attributed to the gradual alignment and stabilization of initially misaligned or weakly polarized ferroelectric domains under repeated electrical cycling. This domain evolution enhances net polarization, a process known as electrical cycling polarization treatment in ferroelectric materials.

While the primary objective of fatigue testing is typically to assess the material’s stability and degradation behavior under millions of electrical cycles, our findings suggest that before any catastrophic failure (e.g., polarization collapse, leakage, or dielectric breakdown) occurs, moderate electric field cycling can actually improve the material’s performance. Under the test conditions used in this study (±10 V, 1000 Hz, and 10^6^ cycles), the Ti/TiO_2_/STO/PZT sample demonstrated excellent polarization stability and fatigue resistance, along with a notable performance enhancement effect. These results highlight the sample’s strong potential for applications in high-frequency actuators and piezoelectric devices, providing a solid foundation for future functional integration.

### 3.6. I–V Characteristics and Leakage Current Analysis

To further investigate the impact of different heterostructures on device insulation performance, current–voltage (I–V) measurements were conducted. In preparation for future device integration, we tested three PZT structures—Ti/PZT, Ti/TiO_2_/PZT, and Ti/TiO_2_/STO/PZT—on both flat and curved substrates, with the curved substrates having a curvature radius equivalent to a circular form with a 50 mm diameter. The I–V measurements effectively reflect the leakage current behavior under an applied DC voltage. Figure 15 and Figure 16 show the I–V characteristics of the three structures on flat and curved surfaces, respectively. The vertical axis represents current on a logarithmic scale, which enhances resolution in the low-current regime.

The results show that the Ti/TiO_2_/STO/PZT structure with an incorporated SrTiO_3_ (STO) buffer layer exhibited the best insulation behavior under both substrate conditions. Its leakage current remained consistently within the nanoampere range (10^−9^ A) across the entire voltage sweep, significantly lower than the values observed for the other two structures without STO. Remarkably, this excellent insulation was preserved even under curved deformation, with no noticeable current spikes, indicating strong mechanical adaptability and electrical stability.

In contrast, the Ti/PZT and Ti/TiO_2_/PZT samples demonstrated relatively poor insulation performance. Particularly under positive bias, leakage currents often increased into the 10^−7^ A range. This issue became even more pronounced on curved substrates, where higher leakage currents were observed. We speculate that the increased mechanical stress induced by bending may exacerbate defect conduction paths along grain boundaries or through localized film defects.

Overall, the incorporation of an STO buffer layer effectively suppressed leakage current and improved the structural resilience to mechanical curvature. To further contextualize our findings, we compared our measured leakage current values with those reported in recent studies. In our Ti/TiO_2_/STO/PZT structure, the leakage current remained within the range of 10^−12^ to 10^−9^ A under ±3 V bias. In contrast, Liu et al. [57] reported leakage currents on the order of 10^−8^ A for sol–gel-derived PZT films on mica substrates under similar voltage conditions, which may result from interface or grain boundary defects in their fabrication process. Yuan et al. [58] also demonstrated that the appropriate selection of bottom electrode materials could reduce leakage current to approximately 10^−9^ A. Compared with these references, our hydrothermally synthesized multilayer-buffered structure exhibits comparable or superior insulation performance, reinforcing the benefits of our dielectric buffer layer design in suppressing leakage paths and enhancing electrical reliability.

### 3.7. Impedance and Phase Angle Analysis

Impedance analysis was performed to evaluate and compare the electrical and electromechanical performance of PZT thin films fabricated via three different processes. The impedance and phase angle (θ) curves were measured for all samples using identical axis scaling for both impedance and phase. The frequency sweep ranged from 100 Hz to 50 MHz, with an applied AC voltage of 1 V and a high-resolution scan comprising 1600 points.

We first analyzed the impedance behavior of the Ti/PZT and Ti/TiO_2_/PZT structures. As shown in Figure 17a,b, both structures exhibit typical piezoelectric resonance responses, but with noticeable differences. Across the entire frequency range, the Ti/TiO_2_/PZT structure consistently exhibited higher impedance values than Ti/PZT, especially in the high-frequency region near the resonance point, where the minimum impedance of Ti/TiO_2_/PZT remained higher than that of Ti/PZT. This suggests that the TiO_2_ interlayer contributes to improved dielectric properties and insulation, leading to enhanced overall electrical impedance characteristics.

In the low-frequency region (around 100 Hz), Ti/TiO_2_/PZT also showed slightly higher impedance (~10^5^ Ω) compared with Ti/PZT, although the difference was marginal. This behavior indicates the TiO_2_ buffer layer may help suppress low-frequency leakage current, thereby improving stability during quasi-static operation. Regarding the phase angle, the Ti/TiO_2_/PZT structure exhibited a sharp transition in θ near the resonance region (~10^6^ Hz), with peak amplitudes approaching ±90°, indicative of strong electromechanical coupling. In contrast, Ti/PZT demonstrated a more gradual phase shift with a lower peak magnitude, reflecting weaker energy conversion efficiency and polarization response. Subsequently, we analyzed the Ti/TiO_2_/STO/PZT structure, with the results shown in Figure 17c. The device exhibited its minimum impedance value of 8.96 Ω at a resonance frequency of 10.19 MHz. Notably, in the low-frequency region, this sample displayed the highest impedance among the three, suggesting superior insulation. Additionally, its phase angle exhibited the most pronounced transition from negative to positive near resonance, further indicating the strongest electromechanical coupling characteristics among the three structures.

In summary, the Ti/TiO_2_/STO/PZT process demonstrated superior impedance stability and piezoelectric coupling efficiency compared with both Ti/PZT and Ti/TiO_2_/PZT structures. These results highlight the beneficial role of the STO buffer layer in improving the overall electrical performance of PZT thin films, confirming its value in advanced piezoelectric device fabrication. This observed improvement in impedance stability and electromechanical coupling for the Ti/TiO_2_/STO/PZT structure can be attributed to the combined effects of improved crystallinity, better lattice matching, and enhanced interface quality provided by the STO buffer layer. As discussed in earlier sections, the STO layer reduces lattice mismatch at the PZT/substrate interface and likely results in residual strain, promoting the formation of highly crystalline MPB-phase PZT with uniform polarization behavior. The improved microstructural integrity and reduced defect density can suppress charge trapping and leakage paths, leading to superior insulation properties, as reflected in the higher low-frequency impedance. Additionally, the better dielectric isolation and interface quality stabilize the electric field distribution across the film, enhancing domain switching dynamics and resulting in the strongest observed electromechanical coupling response. These factors together explain the clear performance advantages of the Ti/TiO_2_/STO/PZT process in impedance and phase angle measurements.

## 4. Conclusions

In this study, we successfully developed high-quality, flexible PZT piezoelectric thin films on titanium foil substrates via a hydrothermal method and systematically investigated the effects of TiO_2_ and SrTiO_3_ (STO) buffer layers on their structural and electrical performance. The multilayered Ti/TiO_2_/STO configuration effectively reduced lattice mismatch and interfacial strain between the PZT film and the metal substrate, facilitating the stable formation of a morphotropic phase boundary (MPB) composition and enhancing crystallinity and microstructural uniformity.

XRD analysis confirmed the presence of the MPB-phase PZT in the STO-buffered structure, while SEM and FIB observations revealed finer grains and smoother film morphology. The Ti/TiO_2_/STO/PZT sample exhibited the highest piezoelectric coefficient (d_33_ = 113.42 pm/V) and superior ferroelectric performance, with a saturated and stable P–E hysteresis loop and a remanent polarization (Pr) of 4.084 μC/cm^2^. Furthermore, the sample demonstrated excellent fatigue endurance, maintaining performance after 10^6^ electrical cycles, and significantly suppressed leakage current, as confirmed by I–V measurements. Impedance analysis further indicated improved electromechanical coupling efficiency in the STO-based heterostructure.

Collectively, the results highlight the strong potential of the Ti/TiO_2_/STO/PZT flexible heterostructure for high-performance applications in actuators, flexible sensors, and advanced microelectromechanical systems (MEMS). Future work may further explore process optimization, long-term mechanical reliability, and integration with practical device platforms to advance its real-world applicability.

## Figures and Tables

**Figure 1 micromachines-16-00879-f001:**
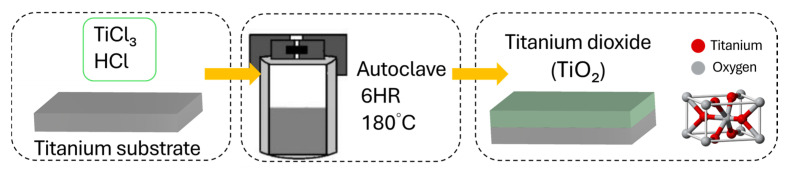
Hydrothermal growth process of the TiO_2_ layer.

**Figure 2 micromachines-16-00879-f002:**
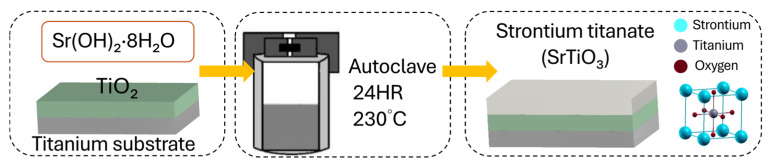
Hydrothermal growth process of the SrTiO_3_ layer.

**Figure 3 micromachines-16-00879-f003:**
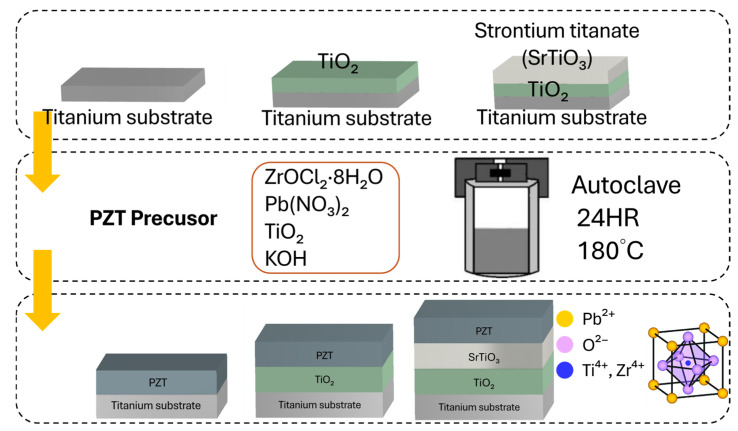
Process flow diagram of PZT growth via the hydrothermal method.

**Figure 4 micromachines-16-00879-f004:**
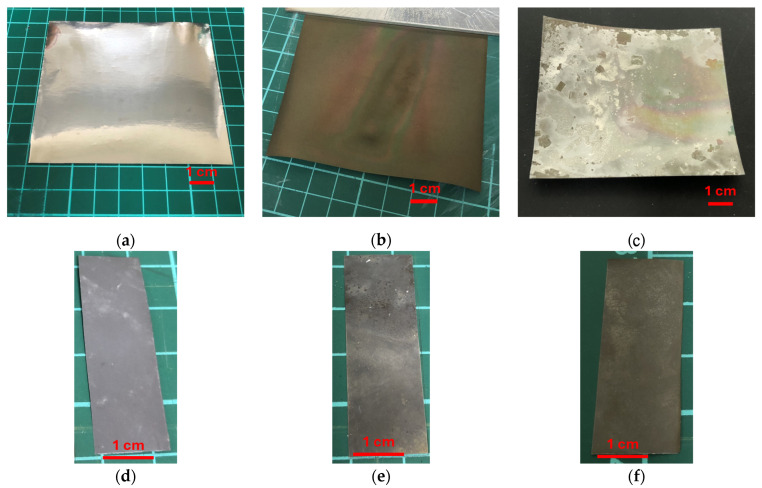
Photographs of flexible titanium substrates and corresponding PZT-coated samples after hydrothermal synthesis. (**a**) Bare titanium foil substrate. (**b**) Titanium substrate after hydrothermal growth of TiO_2_ buffer layer. (**c**) Titanium substrate with sequentially deposited TiO_2_ and SrTiO_3_ (STO) layers. (**d**) Ti/PZT sample synthesized via direct hydrothermal deposition of PZT on bare Ti. (**e**) Ti/TiO_2_/PZT sample with a TiO_2_ buffer layer beneath the PZT film. (**f**) Ti/TiO_2_/STO/PZT sample with both TiO_2_ and STO interlayers introduced prior to PZT growth.

**Figure 5 micromachines-16-00879-f005:**
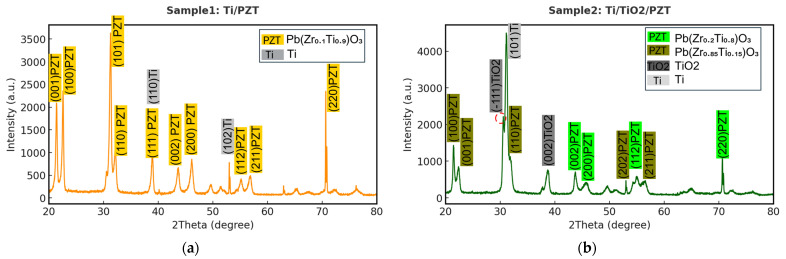
(**a**–**c**) XRD patterns of PZT thin films fabricated with different buffer layers: (**a**) Ti/PZT, (**b**) Ti/TiO_2_/PZT, and (**c**) Ti/TiO_2_/STO/PZT. (**d**) Overlay of XRD patterns highlighting the structural differences among the three samples.

**Figure 6 micromachines-16-00879-f006:**
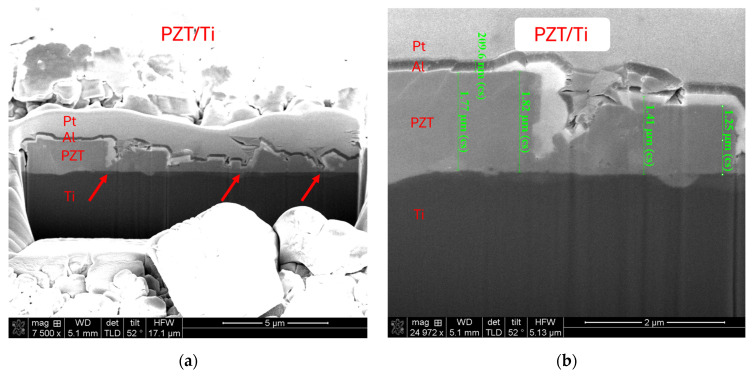
Cross-sectional FIB-SEM images of PZT thin films deposited on titanium substrates with different interlayer structures: (**a**,**b**) Ti/PZT, (**c**,**d**) Ti/TiO_2_/PZT, and (**e**–**h**) Ti/TiO_2_/STO/PZT.

**Figure 7 micromachines-16-00879-f007:**
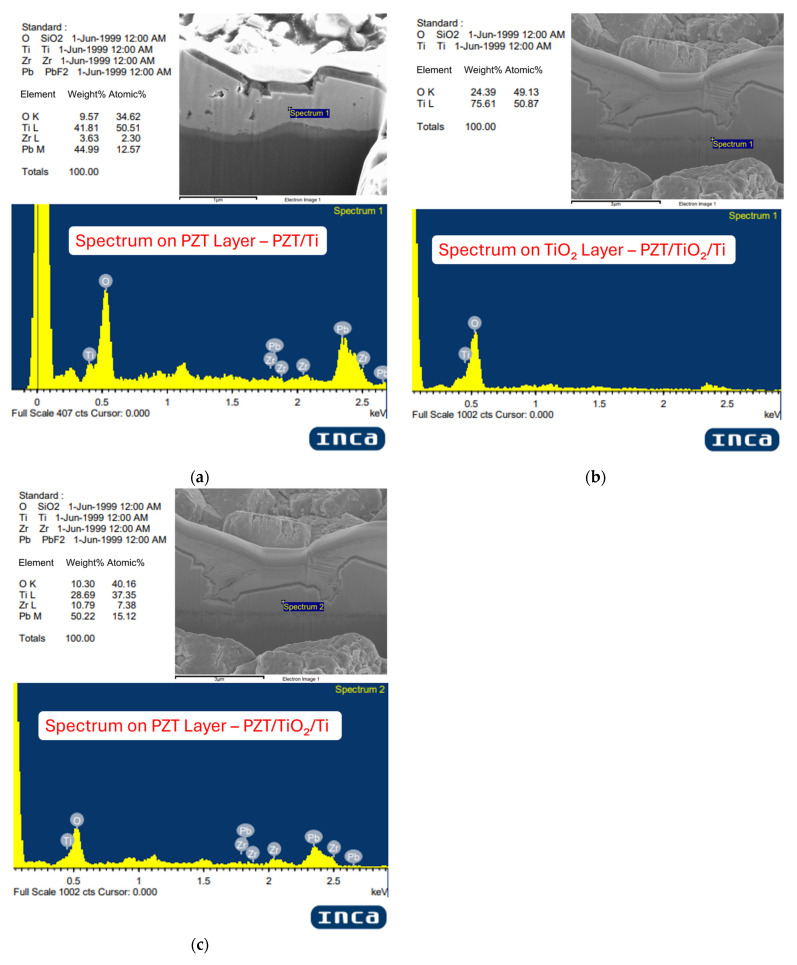
EDX analysis for Ti/PZT and Ti/TiO_2_/PZT samples: (**a**) PZT layer in Ti/PZT, (**b**) TiO_2_ layer in Ti/TiO_2_/PZT, and (**c**) PZT layer in Ti/TiO_2_/PZT.

**Figure 8 micromachines-16-00879-f008:**
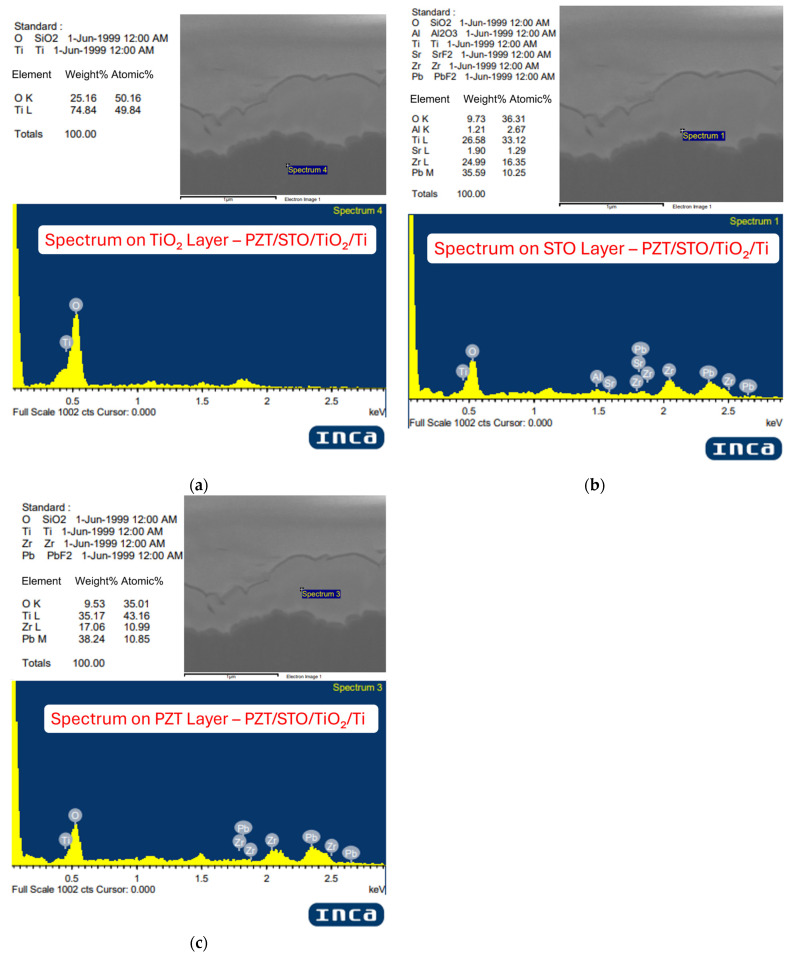
EDX analysis for Ti/TiO_2_/STO/PZT samples: (**a**) TiO_2_ layer in Ti/TiO_2_/STO/PZT, (**b**) STO layer in Ti/TiO_2_/STO/PZT, and (**c**) PZT layer in Ti/TiO_2_/STO/PZT.

**Figure 9 micromachines-16-00879-f009:**
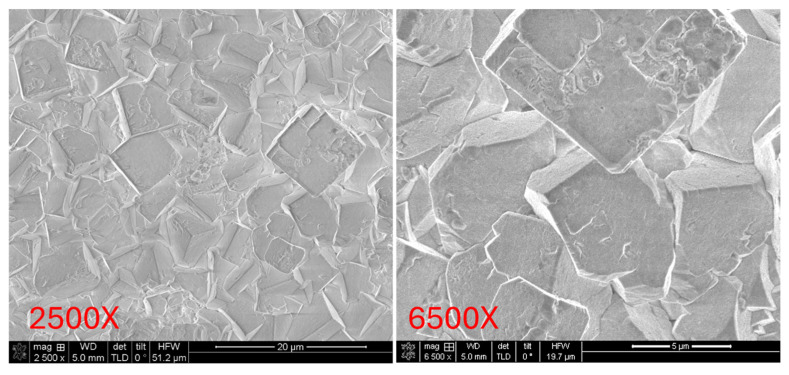
Top-view SEM images of the Ti/PZT sample at 2500× and 6500× magnifications.

**Figure 10 micromachines-16-00879-f010:**
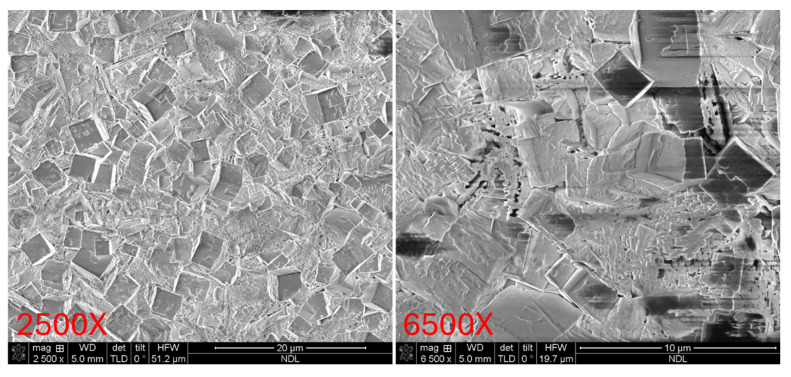
Top-view SEM images of the Ti/TiO_2_/PZT sample at 2500× and 6500× magnifications.

**Figure 11 micromachines-16-00879-f011:**
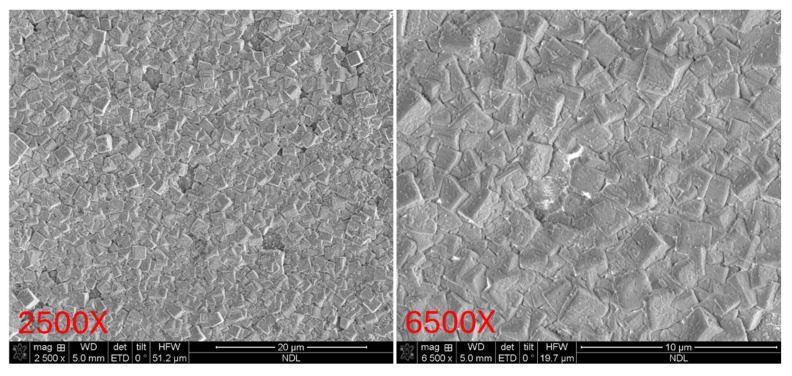
Top-view SEM images of the Ti/TiO_2_/STO/PZT sample at 2500× and 6500× magnifications.

**Figure 12 micromachines-16-00879-f012:**
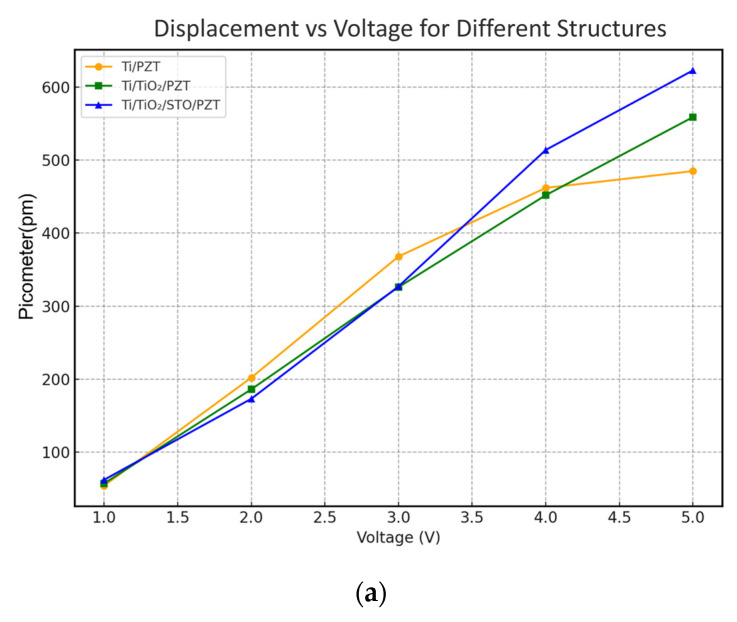
(**a**) Displacement response (in pm) of PZT thin films fabricated by three different processes under applied voltages ranging from 1 to 5 V. (**b**–**d**) Representative PFM amplitude maps at 1–5 V for (**b**) Ti/PZT, (**c**) Ti/TiO_2_/PZT, and (**d**) Ti/TiO_2_/STO/PZT samples.

**Figure 13 micromachines-16-00879-f013:**
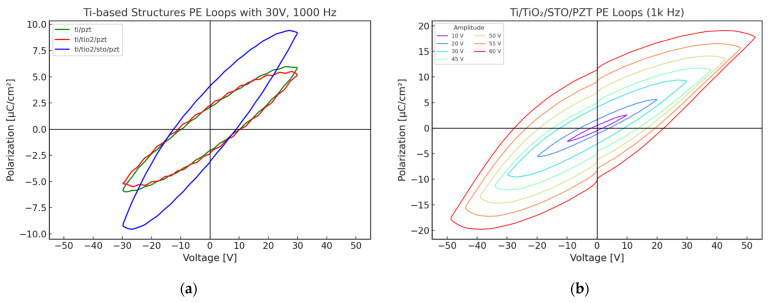
(**a**) P–E hysteresis loops of PZT thin films fabricated by three different processes under 30 V and 1000 Hz; (**b**) P–E hysteresis loops of the Ti/TiO_2_/STO/PZT structure under various applied voltages at 1000 Hz.

**Figure 14 micromachines-16-00879-f014:**
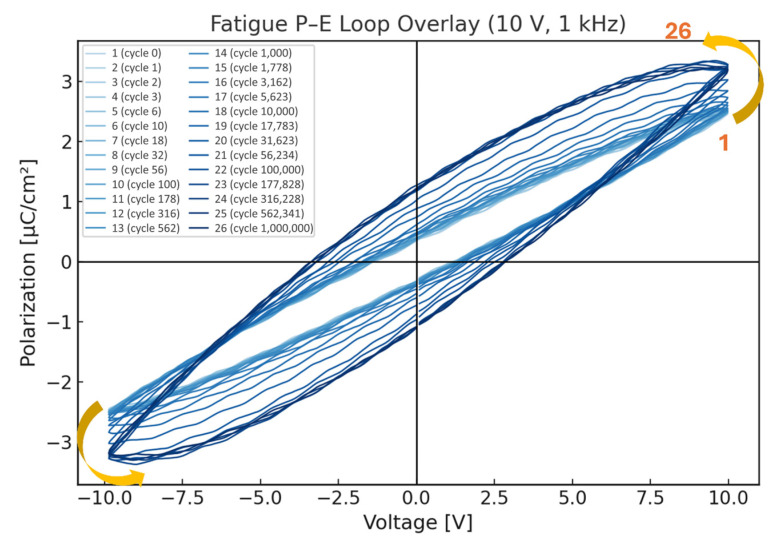
Overlay of P–E hysteresis loops of the Ti/TiO_2_/STO/PZT sample under 10 V fatigue testing at various cycle numbers.

**Figure 15 micromachines-16-00879-f015:**
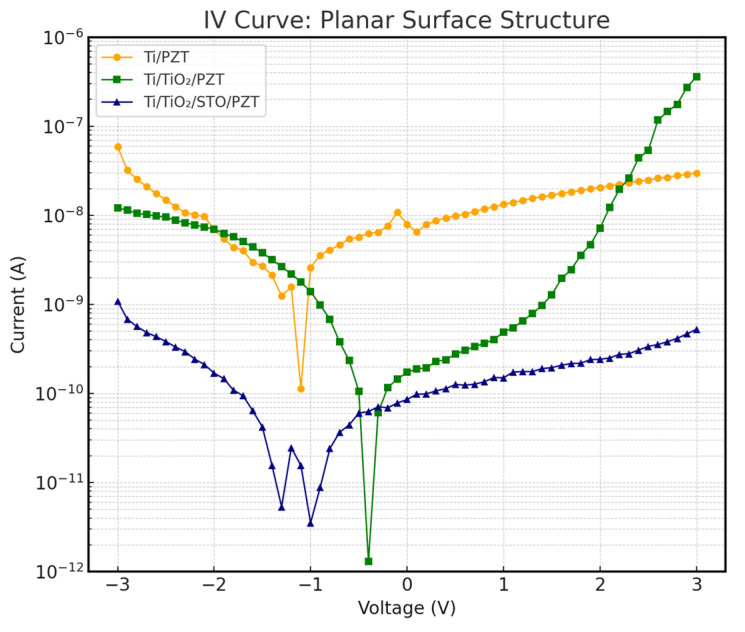
I–V characteristics of the three PZT structures on planar substrates.

**Figure 16 micromachines-16-00879-f016:**
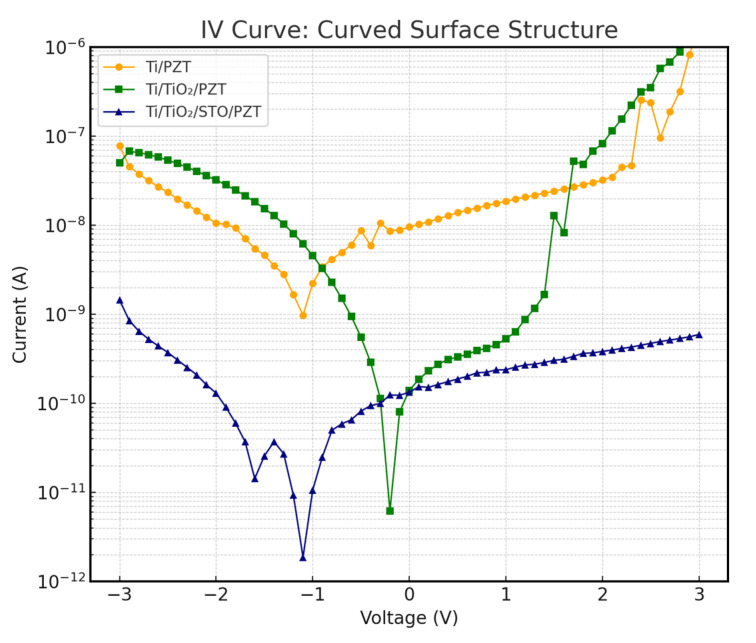
I–V characteristics of the three PZT structures on curved substrates.

**Figure 17 micromachines-16-00879-f017:**
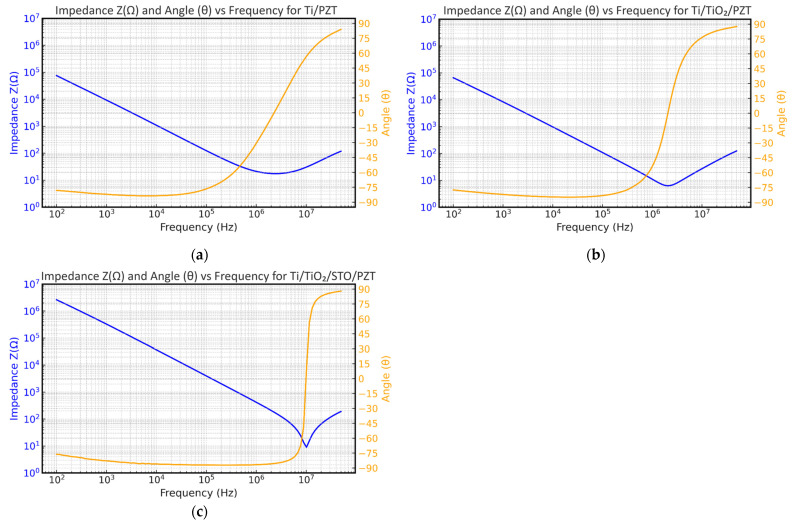
Impedance and phase angle analysis of PZT thin films deposited on titanium substrates with different interlayer structures: (**a**) Ti/PZT, (**b**) Ti/TiO_2_/PZT, and (**c**) Ti/TiO_2_/STO/PZT.

**Table 1 micromachines-16-00879-t001:** Comparison of lattice constants between Ti/TiO_2_ and MPB-phase PZT.

Substrate Material	Lattice Constant a (Å)	Mismatch with PZT (%)	Crystallization Behavior
Titanium foil (α-Ti)	2.951	25%	High mismatch; MPB-phase PZT is difficult to nucleate and grow.
TiO_2_ (Rutile)	4.593	16.7%	Moderate mismatch; may improve crystal growth.

**Table 2 micromachines-16-00879-t002:** Crystal structures and lattice constants of PZT with different phase compositions.

Composition Range	Crystal Structure	a-Axis Range (Å)	Description
Zr-rich (x > 0.6)	Rhombohedral	4.05–4.10	Crystal structure tends toward cubic symmetry.
MPB (x ≈ 0.52)	Tetragonal/Rhombohedral (mixed phase)	≈3.935	Approximate average value.
Ti-rich (x < 0.48)	Tetragonal	3.88–3.91	Elongated c-axis and slightly compressed a-axis.

**Table 3 micromachines-16-00879-t003:** Lattice compatibility calculation between PZT and TiO_2_/STO based on crystal structures and lattice constants.

Substrate Material	Crystal Structure	Lattice Constant a (Å)	Mismatch with PZT (%)	Reference PDF Card
Titanium foil (α-Ti)	HCP	2.951	25%	04-008-1385
TiO_2_	Rutile	4.593	16.7%	01-070-6826
STO	Cubic	3.905	0.76%	04-011-7249
PZT (MPB)	Tetragonal	3.935	-	00-057-0525

**Table 4 micromachines-16-00879-t004:** Summary of XRD peak positions, FWHMs, and estimated crystal sizes of PZT thin films with different fabrication structures.

Structure	Peak Position (2*θ*)	FWHM	Crystal Size, D (nm)
Ti/PZT	31.29°	0.367	224.93
Ti/TiO_2_/PZT	31.10°	0.521	158.07
Ti/TiO_2_/STO/PZT	32.37°	0.703	117.72

**Table 5 micromachines-16-00879-t005:** Displacement response (pm) of PZT thin films fabricated by three different processes under applied voltages from 1 to 5 V.

Structure	1 V	2 V	3 V	4 V	5 V
Ti/PZT	55 pm	202.6 pm	368.7 pm	462 pm	485.6 pm
Ti/TiO_2_/PZT	57 pm	186.1 pm	326.2 pm	425.1 pm	158.1 pm
Ti/TiO_2_/STO/PZT	62.2 pm	173.4 pm	327.3 pm	514.7 pm	623.9 pm

**Table 6 micromachines-16-00879-t006:** Corresponding cycle numbers for each P–E curve of the Ti/TiO_2_/STO/PZT sample during fatigue testing.

Curve Number	Cycle Count	Curve Number	Cycle Count	Curve Number	Cycle Count
1	0	11	178	21	56,234
2	1	12	316	22	100,000
3	2	13	562	23	177,828
4	3	14	1000	24	316,228
5	6	15	1778	25	562,341
6	10	16	3162	26	1,000,000
7	18	17	5623	-	-
8	32	18	10,000	-	-
9	56	19	17,783	-	-
10	100	20	31,623	-	-

## Data Availability

The numerical and experimental data sets generated and analyzed during the current study are available from the corresponding author on reasonable request.

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
