# Peer review of "Hydrothermal Engineering of Ferroelectric PZT Thin Films Tailoring Electrical and Ferroelectric Properties via TiO2 and SrTiO3 Interlayers for Advanced MEMS"

_micromachines, 2025, doi:10.3390/mi16080879_

Round 1
Reviewer 1 Report
Comments and Suggestions for Authors
General comments:
In the article “Hydrothermal Engineering of Flexible Ferroelectric PZT Thin Films Tailoring Electrical and Ferroelectric Properties via TiOâ‚‚ and SrTiO₃ Interlayers for Advanced MEMS, the authors presents an innovative hydrothermal approach for fabricating flexible piezoelectric PZT thin films on 20 μm titanium foil substrates using TiOâ‚‚ and SrTiO₃ (STO) interlayers.
Introduction
The introduction contains too many technical details about deposition techniques and is a tad too long. Think about providing a brief overview of the background and emphasizing the research gap and the distinctive characteristics of your methodology.
Furthermore, there is a lack of clarity regarding the study's purpose and, more crucially, how innovative it is in comparison to previous methods. To more clearly underline the novelty degree, these elements should to be mentioned in the introduction earlier.
Experimental
- Some of the abbreviations used in the experimental section are not clearly reintroduced, making it a little difficult to follow the text smoothly. It would be helpful if key terms were briefly recalled when they are first used in thesection, so that the reader does not have to return to the introduction repeatedly.
- Figure 3 contains a few typographical errors in the chemical formulas, particularly in the subscripts. Please review the figure carefully and correct the formatting to ensure consistency and accuracy.
- XRD structural characterization is well performed and clearly presented.
- For the FIB Cross-Section and EDX elemental analysis, it would be useful to provide more details on the working conditions and procedure used for sample preparation and cross-section measurement. This information is important for understanding the technique used.
- In Figure 6, layers deposited in a sequential order are marked in red, including Al and Pt. However, their role and presence are not clearly explained in the main text. It would be useful to clarify what these layers represent and where in the methodology their deposition is described.
- It would be helpful to indicate the thicknesses of the individual layers in the Ti/TiOâ‚‚/STO/PZTstack. Also, please clarify whether the deposition process was tested for reproducibility , for example, if multiple samples were fabricated and showed consistent results.
- In the SEM-based surface morphology and EDX analysis (Figures 10 and 11), it is not clearly explained how the elemental data for the individual layers were obtained. Since the sample consists of layered thin films (a multilayered sandwich structure), standard EDX performed on the surface will collect the signal from several layers simultaneously. Without cross-section preparation (e.g. by FIB), it is technically difficult to distinguish the contributions of each layer. Please clarify the measurement setup and justify how layer-specific information was extracted.
- The PFM measurements presented are of good quality and reveal a clear enhancement of the piezoelectric response upon inclusion of the STO layer. The authors conclude that the addition of the STO layer leads to an enhancement of 8.65% and 7.62% compared to the Ti/PZT and Ti/TiO₂/PZT structures, respectively. However, this result is not discussed or interpreted in the context of the microstructure, interface quality, or possible domain dynamics. A correlation with the structural and morphological features is necessary to support this observation. The authors should provide a possible explanation for the improved d₃₃ performance.
- The reported improvement in impedance stability and piezoelectric coupling for the Ti/TiOâ‚‚/STO/PZT structure is clearly presented, but not explained. The authors should correlate these results with the structural or morphological features discussed earlier. A brief discussion on the possible role of the STO layer—such as improved crystallinity, interface quality, or dielectric properties—would strengthen the scientific value of this section.
- Please double-check the consistency of terminology throughout the manuscript (e.g., d₃₃ vs. piezoelectric coefficient), and ensure that all abbreviations are clearly introduced particularly in the materials preparation section.
The article is generally well written and includes a comprehensive set of measurements that demonstrate a clear improvement in properties when the STO buffer layer is introduced. However, for readers less familiar with the field, it may be difficult to follow why these enhancements occur, as the manuscript lacks clear explanations correlating the observed improvements with the structure and morphology of the samples. I strongly encourage the authors to provide more concrete interpretations and structure–property correlations to enhance the clarity and scientific integration of the work. Congratulations on the valuable experimental effort.
Reviewer 2 Report
Comments and Suggestions for Authors
Journal: Micromachines
Title: Hydrothermal Engineering of Flexible Ferroelectric PZT Thin Films Tailoring Electrical and Ferroelectric Properties via TiO₂ and SrTiO₃ Interlayers for Advanced MEMS
In this paper, the authors presented PZT thin film grown by Ti, TiO2, STO substrates developed by hydrothermal process to enhance the ferroelectric and electrical properties. The XRD characterization and FWHM evaluation confirmed the expected phases. Furthermore, the PE loops results confirmed the ferroelectric response of the all the prepared samples and presented in detail. Indeed, it is an interesting study, and I would like to recommend it for publication but after the careful minor revision against the raised questions and suggestions given below.
Comments:
- Title of the manuscript is mentioned ‘’flexible’’, while the substrate used is not flexible, so it is suggested to remove the word flexible.
- The abstract is written in very generic way, it should be self-explanatory. It is recommended to highlight the key findings quantitatively in the revised abstract.
- The introduction section is lacking highlighting the research gap by citing latest review reports PE loop results. It is recommended to read and cite the following interesting article emphasized on PE loops results (Journal of the European Ceramic Society 45 (1), 116830).
- It is highly recommended to make the Format of manuscript uniform. Please double check the grammar and English language of the revised manuscript.
- It is highly recommended to provide relevant references in each section for comparisons of the current finding. There is no reference to previous study/finding for comparison.
- Finally, the results present the value for d33 in pm/V; however, it seems that not derived from the PFM images/figures. It is recommended to provide more detailed information to enhance the manuscript interesting.
Round 2
Reviewer 2 Report
Comments and Suggestions for Authors
Please double check all the references again. For example, reference [39] is correct except for the author's names, which need to be corrected.